# C-Reactive Protein Pretreatment-Level Evaluation for Ewing’s Sarcoma Prognosis Assessment—A 15-Year Retrospective Single-Centre Study

**DOI:** 10.3390/cancers14235898

**Published:** 2022-11-29

**Authors:** Sarah Consalvo, Florian Hinterwimmer, Norbert Harrasser, Ulrich Lenze, Georg Matziolis, Rüdiger von Eisenhart-Rothe, Carolin Knebel

**Affiliations:** 1Department of Orthopaedics and Sports Orthopaedic, Klinikum Rechts der Isar, Technical University of Munich, 81675 Munich, Germany; 2Institute for AI and Informatics in Medicine, Technical University of Munich, 81675 Munich, Germany; 3Orthopaedic Department, University Hospital Jena, Campus Eisenberg, 81675 Eisenberg, Germany

**Keywords:** Ewing’s sarcoma, CRP, prognosis, metastasis, local recurrence

## Abstract

**Simple Summary:**

The importance of chronic inflammation in favouring a “cancer-friendly” microenvironment in most types of tumours has been analysed, except for in the case of Ewing’s sarcoma. Ewing’s sarcoma is a highly malignant blue round cell tumour that affects 2.9 in 1,000,000 children worldwide. The aim of this retrospective study was to analyse the role of C-reactive protein (CRP) as a prognostic factor. Serum CRP levels were significantly different in patients with a poorer prognosis and in patients who presented distant metastasis, whereas CRP levels were not significantly different in patients with local recurrence. In Ewing’s sarcoma cases, we believe we can consider a CRP pre-treatment value of >0.5 mg/dL as a sensitive prognostic risk factor indication for distant metastasis and poor prognosis.

**Abstract:**

Background: A pathological/inflamed cellular microenvironment state is an additional risk factor for any cancer type. The importance of a chronic inflammation state in most diffuse types of tumour has already been analysed, except for in Ewing’s sarcoma. It is a highly malignant blue round cell tumour, with 90% of cases occurring in patients aged between 5 and 25 years. Worldwide, 2.9 out of 1,000,000 children per year are affected by this malignancy. The aim of this retrospective study was to analyse the role of C-reactive protein (CRP) as a prognostic factor for Ewing’s sarcomas. Methods: This retrospective study at Klinikum rechts der Isar included 82 patients with a confirmed Ewing’s sarcoma diagnosis treated between 2004 and 2019. Preoperative CRP determination was assessed in mg/dL with a normal value established as below 0.5 mg/dL. Disease-free survival time was calculated as the time between the initial diagnosis and an event such as local recurrence or metastasis. Follow-up status was described as death of disease (DOD), no evidence of disease (NED) or alive with disease (AWD). The exclusion criteria of this study included insufficient laboratory values and a lack of information regarding the follow-up status or non-oncological resection. Results: Serum CRP levels were significantly different in patients with a poorer prognosis (DOD) and in patients who presented distant metastasis (*p* = 0.0016 and *p* = 0.009, respectively), whereas CRP levels were not significantly different in patients with local recurrence (*p* = 0.02). The optimal breakpoint that predicted prognosis was 0.5 mg/dL, with a sensitivity of 0.76 and a specificity of 0.74 (AUC 0.81). Univariate CRP analysis level >0.5 mg/dL revealed a hazard ratio of 9.5 (95% CI 3.5–25.5). Conclusions: In Ewing’s sarcoma cases, we consider a CRP pretreatment value >0.5 mg/dL as a sensitive prognostic risk factor indication for distant metastasis and poor prognosis. Further research with more data is required to determine more sensitive cutoff levels.

## 1. Introduction

Ewing’s sarcoma is a highly malignant blue round cell tumour, with 90% of cases occurring in patients between the age of 5 and 25 years. Worldwide, 2.9 out of 1,000,000 children per year are affected by this malignancy, with a slightly higher incidence in male patients (1.5 male for every −1 female) [1].

The 5-year survival has improved over the years from 40% to 70% because of multidisciplinary treatment strategies using chemotherapy and surgical therapy [2,3,4]. Unfortunately, the recurrence rate, even in initially localised tumours, is 25% [5]. More accurate and reliable factors need to be determined to distinguish high-risk patients. Yet, in a recent SEER database study, tumour size, older age and primary site were established as associated with poor prognosis and the presence of distant metastasis at diagnosis [6,7]. EWING 2008 is the current European protocol in which patients are divided into three groups according to clinical risk factors. The 3-year survival rate changes from 7578% for the standard risk group, to 51–55% for the high-risk group [8,9].

In order to identify more parameters for “high-risk” patients the focus of the International Cancer Society was on analysing the carcinoma microenvironment for the most common cancer entities (e.g., breast or lung cancer) [10,11]. The microenvironment consists of an extracellular matrix, immune-inflammatory cells, blood and lymphatic vascular networks. These cells and this matrix can increasingly develop the function of abnormal tissue and play a crucial role in metastatic spread and growth of malignancies [12]. Indeed, a healthy microenvironment may help to promote protection against tumourigenesis [13,14]. Conversely, the pathological state of that environment can be an additional risk factor for any type of cancer. C-reactive protein (CRP) is one of the proteins of the acute phase response and can be easily measured in routine blood sampling. The purpose of the acute phase reaction is to cause tissue damage locally through a local reaction and simultaneously prevent the damage from spreading too far. They are therefore relevant diagnostic markers of the extent of the body’s response. Acute phase proteins are produced within 24 h; during which time, an increase in their concentration in the blood of 25% is measured [15,16].

Protein concentration can thus increase up to 1000-fold. The prognostic relevance of several inflammatory factors must be analysed. In Ewing’s sarcoma, a high white blood cell (WBC) count is demonstrably associated with decreased event-free survival (EFS) [17]. However, the prognostic role of CRP in patients with Ewing’s sarcoma has not been evaluated. This retrospective study’s aim was to evaluate an accessible, worldwide and inexpensive means of assessing a patient’s risk of distant metastases or reduced life expectancy at diagnosis, and thereby assess a further standardised risk factor for treatment evaluation.

## 2. Methods

This retrospective study included 82 patients who were treated at Klinikum rechts der Isar (Munich, Germany) between 2004 and 2019, and had a confirmed diagnosis of Ewing’s sarcoma.

For all patients, diagnoses were validated through a histopathological examination as the reference standard. The local institutional review and ethics board (Klinikum rechts der Isar, Technical University of Munich) approved this retrospective study (N°48/20S). The exclusion criteria included insufficient laboratory values, a lack of information regarding follow-up and Rx or R1/R2 resection status. In total, 40 patients were included in this study. 

### 2.1. Preevaluation

We analysed the medical records of all patients enrolled in this study. Histology was obtained via either incision or CT scan-guided biopsy performed in our institution, and confirmed at an interdisciplinary sarcoma board following the WHO guidelines.

Laboratory data were collected during pretreatment between one and up to a maximum of seven days before biopsy or the first surgical treatment at the first appointment at our clinic. All patients were screened for bacterial infection with urinary samples, lung X-ray and total body clinical examination (e.g., urinary infection, pulmonary disease or other possible systemic bacterial infections) and were excluded if the screenings were positive. CRP levels were reported in milligrams per decilitre (mg/dL) with a normal value considered as below 0.5 mg/dL. The measurement was performed using the Cobas^®^ 8000 modular analyser C702 (Fa. Roche, Basel, Switzerland). The follow-up investigations were performed in line with EWING 2008, International Guideline Harmonisation Group for Late Effects of Childhood Cancer, Late Effects Surveillance System [18].

All patients were followed up in our department every 3 months for the first 2 years, every 6 months from the third to the fifth year and at 12-month intervals thereafter in accordance with guidelines. Disease-free survival (DFS) time was calculated as the time between the initial diagnosis and an event such as local recurrence or distant metastasis. All metastasis locations were included. The follow-up time was defined as the time between the initial diagnosis and the last follow-up, or the last (unscheduled) presentation of the patient for a check-up in our clinic. The follow-up status of patients was classified as death of disease (DOD), no evidence of disease (NED) or, for patients who are currently alive but had a diagnostically confirmed distant metastasis or a proven local recurrence, or alive with disease (AWD).

### 2.2. Statistical Analysis

Data were processed and analysed using StatPlus:mac Pro 2020 (Fa. AnalystSoft, Walnut, CA, USA).

The national standard of 0.5 mg/dL was used to distinguish between ‘increased’ and ‘decreased’ CRP. The correlation between CRP value and overall survival (OS) was explored through a Pearson correlation model. The following were selected as guide values for the interpretation of the correlation coefficient: CC = 0 no linear correlation, CC = 0.3 weak positive linear correlation and CC = 0.5 positive linear correlation.

Statistically significant values were calculated with a *p*-value < 0.01. Sensibility and specificity based on the optimal identified cut point were calculated along the 95% interval.

The survival curves were generated using Kaplan–Meier analysis and evaluated using the log-rank test. The quantitative accuracy of the CRP measurements was measured by the area under the curve (ROC Curve/AUC). An AUC value >0.8 was considered a good predictive test ability. Two sample *t*-tests were performed to estimate the association between CRP levels and prognosis status, the presence of distance metastasis and local recurrence.

## 3. Results

A total of, 82 patients with a confirmed diagnosis of Ewing’s sarcoma were included. In total, 42 patients were excluded: 14 patients presented insufficient laboratory values, 26 cases had a lack of information regarding the follow-up status and 2 cases had non-oncological resection (R1/R2). The inclusion criteria were met by 40 patients. Their clinical characteristics are summarised in Table 1. The mean age of the patients was 21.5 years with a minimum age of 4 years and a maximum age of 62 years. The majority of the cohort was male (70% vs. 30%). The mean female age was 23 years (12–62 years), whereas that of males was 21.5 years (4–54 years). With regard to sarcoma position, 47.5% of patients were affected at lower extremity localisations (27.5% femur and 20% tibia), pelvis 20%, humerus 5%, clavicle 5%, forearm 5%, foot 5% and spine 2.5%. At initial diagnosis, 10% (*n* = 4) showed multifocal bone involvement.

The follow-up status was defined as DOD in 17 patients (42%), NED in 21 patients (53%) and AWD in 2 patients (5%). No surgical therapy was performed in 13 patients (32.5%). In all other patients (67.5%), R0 resection was confirmed by the pathology department (*n* = 27). Of the 13 patients who had no surgical treatment, 11 had proven multiple metastases in CT staging (CT thorax/abdomen/pelvis). Chemotherapy and local radiotherapy were performed in 73% of the aforementioned cases. Palliative radiotherapy was performed on only one patient based on his/her request and in 18% only systemic therapy was performed. No surgical therapy was performed despite the absence of distant metastasis in two patients. In one case, the primary tumour was localised in the distal tibia with a skip lesion in the proximal fibula in the first patient. An oncological resection would have caused the complete loss of foot function with necessary tibial nerve resection. In agreement with the parents, surgery was not performed and only chemotherapy with local radiation was conducted. The other case had a localised Ewing’s sarcoma in the calcaneus and after completing neoadjuvant chemotherapy, he/she demonstrated completely regressive radiological findings, so in agreement with the parents no surgical resection was performed to avoid a functionally mutilating calcanectomy.

The median CRP value of the entire cohort was 1.6 mg/dL (0.1–32.8 mg/dL). 

In the deceased patients, the median CRP level was 4.6 mg/dL (0.1–32.8 mg/dL). In those patients who remained alive but showed distant metastasis or local recurrence, the median CRP level was 0.6 mg/dL (0.1–0.6 mg/dL), whereas the median CRP value of the NED group was 0.7 mg/dL (0.1–5.8 mg/dL) (Figure 1). It can be shown that those patients in our cohort who ultimately died as a result of Ewing’s sarcoma had higher preoperative CRP values on average on the basis of the aforementioned values.

In the group with an elevated CRP (>0.5 mg/dL) value, 59% had already died, 3% was alive but with metastases or a local recurrence and 37% was alive without any disease manifestation. In the low-CRP-value group (<0.5 mg/dL), 8% had already died, 8% were alive but with metastases or a local recurrence and 84% were alive without any manifestation of the disease. Of the entire cohort, 37% had pulmonary metastases, 52% had multifocal metastases (more than two organs) and 10% had metastases in organs other than the lungs. Metastases were already present in 37.5% of patients at initial diagnosis. 

### Survival Curves

The median follow-up time was 4.8 years (0–17 years). In contrast, the median DFS time was 3.5 years (0–17 years). Different survival curves could be established on the basis of the predefined cutoff value of 0.5 mg/dL depending on the CRP value.

Patients with a preoperative CRP value of >0.5 mg/dL had a median follow-up time of 4.5 years (2 months–13 years). In contrast, patients with a CRP value below the cutoff value of 0.5 mg/dL showed a median follow-up time of 5 years (0–17 years), as shown in Figure 2.

Univariate analysis of CRP level > 0.5 mg/dL revealed a hazard ratio (HR) of 9.5 (95% CI 3.5 to 25.5). A 5-year survival rate of 92% and 46% was calculated in the CRP < 0.5 mg/dL and >0.5 mg/dL groups, respectively.

The Ewing cohort showed a median DFS time of 7 months (0–17 years) with a preoperative CRP above 0.5 mg/dL, whereas patients with a CRP below 0.5 mg/dL showed a median DFS time of 5.5 years (0–11 years), as shown in Figure 3.

The risk estimation of the recurrence of an event in terms of local recurrence or distant metastasis was 8.3 times higher in the patients with an CRP level > 0.5 mg/dL compared to the group of patients with a CRP level below 0.5 mg/dL (8.3 HR and 95% CI 3–22.7).

The optimal breakpoint was confirmed to be 0.5 mg/dL (AUC 0.812, sensitivity = 0.76 and specificity = 0.74) (Figure 4).

The correlation between the CRP level and a reduced DFS as well as poor prognosis in Ewing sarcomas showed a CC (Correlation Coefficient) of 0.51, with a *p*-value < 0.0005.

Two sample *t*-tests confirmed that CRP levels were a significant prognostic factor for poor prognosis (*p* < 0.001) and risk of the presence of distant metastasis (*p* = 0.009). No significantly relevant difference was found between CRP level and occurrence or local recurrence (*p* = 0.02) (Table 2).

## 4. Discussion

This study aimed to determine the prognostic value of a pretreatment serum CRP analysis. After evaluating the statistical relevance of CRP values in correlation with prognosis, we can prudently confirm the involvement of this protein in the acute phase response in the Ewing sarcoma microenvironment.

In accordance with our findings, Aggerholm-Pendersen et al. analysed a group of 172 patients with bone sarcoma (consisting of 63 chondrosarcomas and 109 between Ewing’s sarcomas and osteosarcomas) and demonstrated that elevated CRP levels were associated with increased overall mortality [19]. In contrast to our study, osteosarcomas and Ewing’s sarcomas were analysed in one group without any entity distinction. Likewise, in 2013, a poorer DFS was statistically confirmed (*p* = 0.02) in patients with Ewing’s sarcoma and chondrosarcoma with an elevated CRP value. Here, no cutoff value was calculated and no distinction per entity was performed. Furthermore, patients with metastatic spread at diagnosis were excluded [20]. The calculated 5-year survival with elevated and reduced CRP levels was 57% and 79% (*p* < 0.0001), respectively [20]. In our study, we calculated a 5-year survival of 46% and 92% in the elevated and reduced CRP level groups (*p* = 0.006), respectively. Survival was not completely comparable because of the inclusion of chondrosarcomas in the Nakamura et al. study.

Two recent studies analysed the CRP–albumin ratio (CAR). In a cohort of 122 Ewing’s sarcomas, Yong-Jiang et al. demonstrated that the CAR had a significantly larger AUC compared with the neutrophil–leucocyte ratio (NLR), platelet–leucocyte ratio, leucocyte–monocyte ratio and neutrophil–platelet ratio ROC curves. Therefore, higher levels of CAR were correlated with poor prognosis (HR 2.4, *p* = 0.005), and the calculated ratio was the most robust prognostic factor of all the aforementioned factors [21]. An optimal CAR cutoff value of 1.5 as a prognostic factor was calculated two years later. Therefore, the presence of metastasis and a CAR value under <1.5 were significantly associated with a reduced OS (*p* < 0.05). In particular, this study was performed only on spinal Ewing’s sarcomas [22]. Even if the analysed prognostic factor was a ratio (CRP/albumin), a correlation and confirmation of our findings can be made.

Considering other inflammation agents, Biswas et al. predicted inferior EFS (*p* = 0.009) and local control (*p* = 0.02) rates for patients without metastases in a cohort of 60 extraosseous Ewing’s sarcomas with a WBC count of >11 × 10^9^/L [23].

Pretreatment levels of 224 localised Ewing’s sarcomas were analysed and a count of >11 × 10^9^/L WBC predicted an inferior EFS (*p* = 0.003) [24]. In the second study, 35 cases of head and neck Ewing’s sarcoma were analysed that were diagnosed in the same institution and were treated with a uniform chemotherapy protocol. Multivariate analysis showed that baseline the WBC count independently predicted the EFS rate (*p* = 0.04). Patients with WBC ≤ 11.000/μL had superior EFS, although no difference for OS was observed [25]. Although only for osteosarcomas, there are currently encouraging results regarding the better prognosis of these patients with addition of the conventional chemotherapy of mifamurtide. This is an immunomodulator that stimulates the immune response against tumour cells, e.g., in the lung, by activating macrophages and releasing proinflammatory or antitumour cytokines. This represents clear evidence of the immune system’s central role in carcinogenesis even if it is currently only for osteosarcoma. The role of tumour-associated macrophages (TAMs) is also becoming a central question in the OS (Overall Survival) of cancer entities. Fujiwara et al. found that a high extent of TAM infiltration, a substantial microvascular density, elevated WBC counts (>6800 cells/μL) and CRP values > 0.2 mg/dL were significantly associated with worse prognosis [26]. In these cases, the CRP cut-off was set lower than in our study.

In consideration of our findings, we can confirm the role of pretreatment CRP values in prognosis, local recurrence and of the presence of distant metastasis in patients with Ewing’s sarcoma (Table 2). The advantages of CRP blood testing over the above options are its lower cost and ubiquitous use in any clinic or outpatient practice. Sadly, it is very difficult to clearly confirm the clinical implication of the findings because of the rarity of this cancer subtype, and the small size of the analysed cohort. Even with a small sample size, we think that a confirmed R0 resection of Ewing’s sarcoma can help eliminate an important bias in local recurrence and distant metastasis. Unfortunately, numerous patients were lost in the 15-year follow-up range, and complete follow-up could not be calculated.

### Mifamurtide

We would like to put some focus on the new drug mifamurtide, to better understand the clinical implications of chronic inflammation regarding the “cancer-friendly” state.

Mifamurtide/L-MTP-PE (liposomal muramyl tripeptide phosphatidylethanolamine) or MAPACT (Takeda Pharmaceutical Company) is a synthetic drug which stimulates the immune response, thereby activating macrophages and monocytes [27]. MAPACT is currently approved in Europe for the treatment of non-metastatic Osteosarcoma in addition to standard chemotherapy [28,29]. 

Studies in vitro demonstrated that human macrophages can be induced by Mifamurtide to stimulate anti-tumor activity against Osteosarcoma cells. Punzo et al. treated macrophages obtained from peripheral blood mononucleated cells of healthy donors and MG63 cells (cells that have fibroblast morphology and are isolated from the bone patient with osteosarcoma) with Mifamurtide. MG63 cells co-cultured with Mifamurtide-activated macrophages showed a significant decrease in the metastasis, prognosis and inflammation markers compared to those co-cultured with macrophages which were not activated [30].

Mifamurtide is pharmacological proof that an immunomodulatory drug given together with standard adjuvant chemotherapy can improve the prognosis of cancer patients.

## 5. Conclusions

It can be confirmed that a preoperative evaluation of the CRP value in patients with Ewing’s sarcoma represents a valuable prognosis indicator. Thus, the follow-up for patients with elevated preoperative CRP value >0.5 mg/dL could be modified for this higher-risk group. Further research and the central collection and analysis of data is required to determine the most sensitive cutoff values of pretreatment CRP levels in predicting poor survival or distant metastasis.

## Figures and Tables

**Figure 1 cancers-14-05898-f001:**
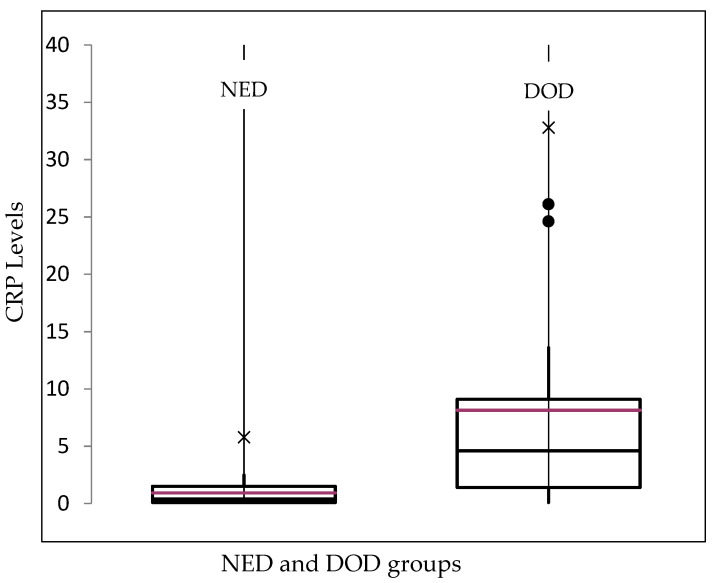
Box plot of the median CRP values (red line) depending on the follow-up status of the patients: no evidence of disease (NED) and death of disease (DOD). For the alive with disease (AWD) status, no diagram was created because only two patients were involved. ×: extreme outliners, ●: mild outliners.

**Figure 2 cancers-14-05898-f002:**
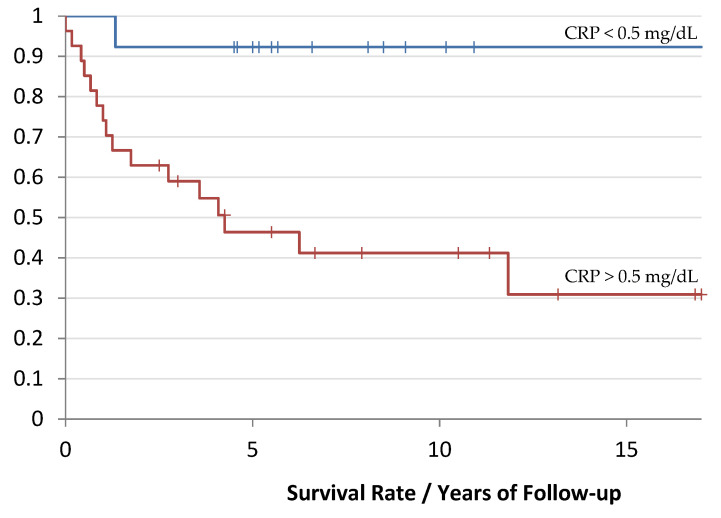
Kaplan-Meier curve of the survival rate depending on the CRP value: red corresponds to the group with an increased CRP value and blue corresponds to the group with a lower CRP value. The follow-up time is evaluated in years (*p*-value: 0.005).

**Figure 3 cancers-14-05898-f003:**
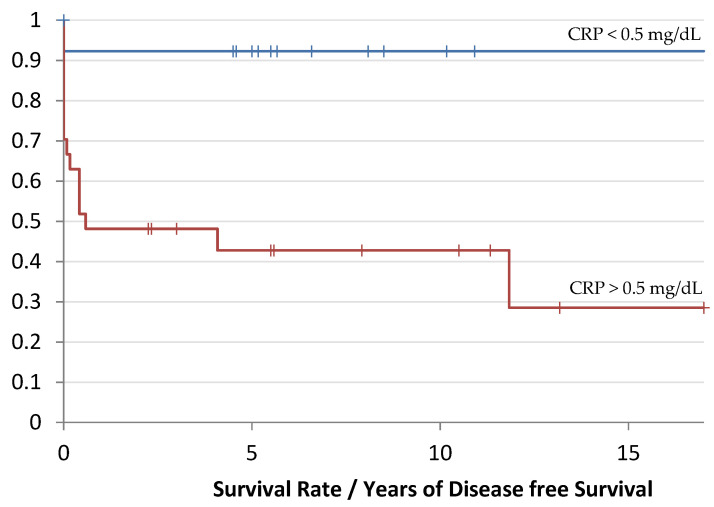
Kaplan-Meier curve of the recurrence/remote metastasis-free time as a function of the CRP value: red corresponds to the group with an increased CRP value andblue corresponds to the group with a lower CRP value. The DFS time is evaluated in years (*p*-value: 0.006).

**Figure 4 cancers-14-05898-f004:**
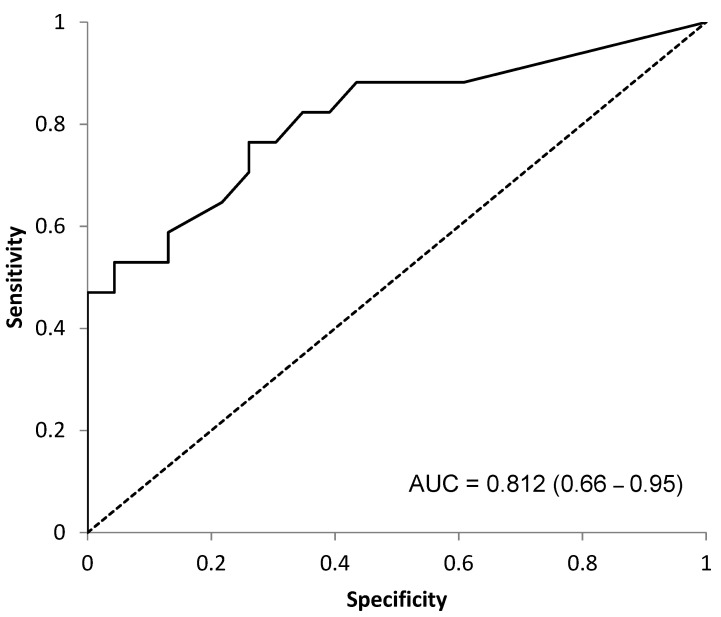
ROC Curve (AUC = 0.812). Optimal breakpoint analysis of CRP as a prognostic factor of OS.

**Table 1 cancers-14-05898-t001:** Clinical characteristics and descriptive statistics of cohort.

Spalte1	*n*	Percentage	Mean	Range	SD
Age	40		21.5 years	4–62 years	13.54 years
Sex					
Male	28	70%	-	-	-
Female	12	30%	-	-	-
Site					
Femur	11	27.50%	-	-	-
Tibia	8	20%	-	-	-
Humerus	2	5%	-	-	-
Clavicula	2	5%	-	-	-
Radius	2	5%	-	-	-
Hip	8	20%	-	-	-
Foot	2	5%	-	-	-
Spine	1	2.50%	-	-	-
Multifocal	4	10%	-	-	-
Follow-up					
DOD	17	42.50%	-	-	-
NED	21	52.50%	-	-	-
AWD	2	5%	-	-	-
Metastasis					
Pulmonary	7	17.50%	-	-	-
Skeletal	2	5%	-	-	-
Multifocal	10	25%	-	-	-
At diagnosis	15	37.50%	-	-	-
None	6	15%			
Local recurrence	6	15%	-	-	-
DFS	40	-	3.5 years	0–17 years	54.5 years
Follow-up	40	-	4.79 years	0–17 years	53.6 years
CRP					
>0.5 mg/dL	27	67.50%	1.6 mg/dL	0.6–32.8 mg/dL	9 mg/dL
<0.5 md/dL	13	32. 50%	0.1 mg/dL	0.1–0.2 mg/dL	0.05 mg/dL

**Table 2 cancers-14-05898-t002:** Two sample *t*-tests of CRP level as a predictor of the presence of metastasis, local recurrence and poor prognosis.

	CRP Levels		CRP Levels		*p*-Value
	*n* (%) 95% CI		*n* (%) 95% CI		
Presence of			Absence of		
Metastasis	19 (47.5%)	2.3 to 11.9	21 (52.5%)	0.3 to 1.8	0.009
LR	9 (22.5%)	−0.2 to 17.7	31 (77.5%)	0.6 to 4.5	0.02
Prognosis	DOD		NED/AWD		
	17 (42.5%)	2.9 to 13.3	23 (57.5%)	0.3 to 1.4	<0.001

## Data Availability

The data that support the findings of this study are available on request from the corresponding author (Dott. S. Consalvo). The data are not publicly available due to restrictions e.g., their containing information that could compromise the privacy of research participants.

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
