# Peer review of "C-Reactive Protein Pretreatment-Level Evaluation for Ewing’s Sarcoma Prognosis Assessment—A 15-Year Retrospective Single-Centre Study"

_cancers, 2022, doi:10.3390/cancers14235898_

Round 1
Reviewer 1 Report
This study by Consalvo, et. al., is a retrospective analysis of whether or not C-reactive protein levels (CRP) are a useful prognostic marker for Ewing sarcoma patients. The rationale for this study is that CRP is a marker for the inflammatory environment that promotes metastasis, is readily and cheaply detectable, and could be used to identify high risk patients at diagnosis. In general, the sample size is small, but supports the hypothesis and is in line with previous studies. This is a rare tumor so the small sample size is not surprising and the authors discuss the need for an expanded cohort.
This article is appropriate for publication in Cancers, but requires editing of the text. The writing, particularly in the discussion, is somewhat choppy and contains incomplete sentences. There paragraph breaks seem to be inappropriate in several places in the discussion. Perhaps a native English editor is needed. There are also some small, but significant errors in the text that need to be corrected prior to publication. Find my point-by-point comments below:
1) In lines 112-115, there is a methodology discussed regarding using a Pearson correlation to evaluate the correlation between overall survival and CRP. The results of this analysis do not appear to be reported in the manuscript, so the authors must add that analysis or remove this section of the methods.
2) The authors use the word "collective" throughout. A more appropriate word might be "cohort."
3) There should be statistical measures of significance reported with Figures 1, 2, and 3.
4) For line 184, it seems as though the numbers for either the 5-year overall survival rate or the CRP values need to be reversed in order for the paper to support the authors hypothesis. As written, this suggests that 5-year survival is better in patients with higher CRP.
5) Line 193 is currently a stand-alone paragraph. This does not seem correct.
6) In the sentence in line 194, it is not clear what the 8.3 HR is for. What is there an increased risk of?
7) The sentences on lines 195-197 are heavily redundant and should be revised for concision.
8) Table 2 is confusing. The 95% CI reported look like odds ratios, but this is not clear from the legends. Also, it's not clear what the odds ratio is in reference to?
9) The run-on sentence on lines 209-212 is a word salad, with an unclear meaning. This needs to be revised for clarity.
10) The sentence on lines 214-216 is not a coherent sentence.
11) Should the sentence on line 226 be included in the paragraph on line 228?
12) Is there an exponent is missing in the "11 x 10/L" on lines 241 and 243? Especially with a WBC of 11,000/uL referenced later, there are still 3 orders of magnitude unaccounted for.
13) Should the sentence on line 239 be included in the paragraph on line 242?
14) There is insufficient context leading to the discussion about mifamurtide. This may need a separate paragraph.
15) The sentence on line 256 should be included with a paragraph and not standing alone.
Author Response
We would like to thank you for your thorough review and thoughtful comments. We have modified and revised the resubmitted manuscript according to the comments. Please find our point-by-point responses below.
-In lines 112-115, there is a methodology discussed regarding using a Pearson correlation to evaluate the correlation between overall survival and CRP. The results of this analysis do not appear to be reported in the manuscript, so the authors must add that analysis or remove this section of the methods.
Pearson correlation added at line 201-202
-The authors use the word "collective" throughout. A more appropriate word might be "cohort."
Adjusted accordingly.
-There should be statistical measures of significance reported with Figures 1, 2, and 3.
Adjusted accordingly for Figure 2 and 3. Figure 1 is a Box Plot, so no other possible statistical measure is possible. the max/min are in the text
-For line 184, it seems as though the numbers for either the 5-year overall survival rate or the CRP values need to be reversed in order for the paper to support the authors hypothesis. As written, this suggests that 5-year survival is better in patients with higher CRP.
Adjusted accordingly.
- Line 193 is currently a stand-alone paragraph. This does not seem correct.
Adjusted accordingly.
-In the sentence in line 194, it is not clear what the 8.3 HR is for. What is there an increased risk of?
Explained in the text accordingly.
-The sentences on lines 195-197 are heavily redundant and should be revised for concision.
Adjusted accordingly.
- Table 2 is confusing. The 95% CI reported look like odds ratios, but this is not clear from the legends. Also, it's not clear what the odds ratio is in reference to?
Adjusted accordingly. Table is modified for more clarity.
- The run-on sentence on lines 209-212 is a word salad, with an unclear meaning. This needs to be revised for clarity.
Adjusted accordingly.
- The sentence on lines 214-216 is not a coherent sentence.
Adjusted accordingly.
-Should the sentence on line 226 be included in the paragraph on line 228?
Adjusted accordingly.
- Is there an exponent is missing in the "11 x 10/L" on lines 241 and 243? Especially with a WBC of 11,000/uL referenced later, there are still 3 orders of magnitude unaccounted for.
Adjusted accordingly.
-Should the sentence on line 239 be included in the paragraph on line 242?
Adjusted accordingly.
-There is insufficient context leading to the discussion about mifamurtide. This may need a separate paragraph.
Adjusted accordingly. Separate paragraph added.
-The sentence on line 256 should be included with a paragraph and not standing alone.
Adjusted accordingly.
To conclude, we feel that the paper significantly improved due to your input. We thank you again for your effort. We hope that our responses are considered satisfactory.
For further questions, we remain at your disposal.
Sincerely,
Dr. S.Consalvo
Reviewer 2 Report
Consalvo et al. describe a single institution study to correlate preoperative CRP levels with clinical outcomes in Ewing sarcoma patients. The study was well-designed, described and presented. It has potential clinical relevance.
A non-obligatory comment on the data is the correlation with other parameters, if present, that was shown or investigated by others as described and discussed in the paper, these would include CAS, NLR, WBC and non-discussed parameters like LDH levels. These could be then tested using ANOVA or other multiparametric comparisons to better understand outliers in the cohort. However, it should be discussed why these comparisons were not done.
Minor issues: Please check hyphenation in the text, for example, "nononcological" to replace with "non-oncological", check the text through for similar issues.
Use "." instead of "," as a decimal separator, see, for example, table 1 and use the proper 1000 separator "of 1,000.000 children" from lines 15 and 26.
Line 125 results section: The use of the word "Approximately" for an exact number of 82 patients is not correct, the same stands for line 134 with 4 cases in 40 samples 10% are exact and not approximate.
"Ewing's sarcoma" and "primitive neuroectodermal tumours" lines 13 and 25 are not acceptable nomenclature according to the WHO Classification of tumours 5th edition. See and correct other places in the text, e.g.: lines 19 and 28. Also, use a reference to the proper chapter in this book.
Abbreviations are listed two times in the reviewed format.
Author Response
We would like to thank you for your thorough review and thoughtful comments. We have modified and revised the resubmitted manuscript according to the comments. Please find our point-by-point responses below.
-A non-obligatory comment on the data is the correlation with other parameters, if present, that was shown or investigated by others as described and discussed in the paper, these would include CAS, NLR, WBC and non-discussed parameters like LDH levels. These could be then tested using ANOVA or other multiparametric comparisons to better understand outliers in the cohort. However, it should be discussed why these comparisons were not done.
Thank you for pointing this out.Unfortunately our data regarding NLR,WBC and LDH were incomplete to perform a multiparametric comparison. We are actually integrating these data for other Sarcoma patients (e.g. Chondrosarma, Osteosarkoma) to present a statistical comparison in our next paper.
- Please check hyphenation in the text, for example, "nononcological" to replace with "non-oncological", check the text through for similar issues.
Adjusted accordingly.
-Use "." instead of "," as a decimal separator, see, for example, table 1 and use the proper 1000 separator "of 1,000.000 children" from lines 15 and 26.
Adjusted accordingly.
-Line 125 results section: The use of the word "Approximately" for an exact number of 82 patients is not correct, the same stands for line 134 with 4 cases in 40 samples 10% are exact and not approximate.
Adjusted accordingly.
-"Ewing's sarcoma" and "primitive neuroectodermal tumours" lines 13 and 25 are not acceptable nomenclature according to the WHO Classification of tumours 5th edition. See and correct other places in the text, e.g.: lines 19 and 28. Also, use a reference to the proper chapter in this book.
Adjusted accordingly.
To conclude, we feel that the paper significantly improved due to your input. We thank you again for your effort. We hope that our responses are considered satisfactory.
For further questions, we remain at your disposal.
Sincerely,
Dr. S.Consalvo
Reviewer 3 Report
Title: C‐Reactive Protein Pretreatment-Level Evaluation for Ewing's Sarcoma Prognosis Assessment—A 15-Year Retrospective Single-Centre Study
This article describes a 15-year retrospective study of 82 patients (42 of which were excluded) of diagnosed Ewing sarcoma (ES) at Klinikum rechts der Isar between 2004 and 2019. The authors evaluate the potential of C-reaction protein (CRP) as a prognostic factor for ES patients. CRP serum levels were evaluated in patients 1-7 days prior to biopsy or first surgical treatment. The results of this study showed a statistically significant difference in CRP levels in patients with poorer prognosis and in patients who presented with distant metastasis, but not in patients with local recurrence, suggesting that CRP pretreatment level may have value as a prognostic factor for ES. Overall, this study is well designed and the results are presented very clearly. These results are valuable to the field of Ewing sarcoma research and should be published after a very minor issues are addressed.
1. Simple summary is very similar to abstract. Could be re-worded and simplified for general accessibility.
2. In simple summary and abstract, “2.9 out of 1,000.000” is written. This should be changed to “2.9 out of 1,000,000” for clarity and consistency with the rest of the article.
3. Page 2, line 73. “Because of the relationship between inflammation and cancer” is not a complete sentence and should be reworded.
4. Page 3, lines 119-120. “… was measured By the area under the curve…”. There appears to be an extra space in the sentence and “By” should not be capitalized.
5. Page 10, line 269. “In view results of this paper,…” should be reworded for clarity.
6. In Figure 1, the box plot is labelled “DO”. It should be labelled “DOD” for clarity and consistency.
7. Figures 1-3 are not labelled well. Y-axes are not labelled and the quality of the graphs is relatively low, but readable. All the information is contained in the legends, but it would be better if the graphs could be easily interpreted without having to read the legend.
8. Some sentences/paragraphs should be combined for clarity. Single sentence paragraphs should be combined. For example, the conclusion could probably be combined into a single paragraph.
9. CRP levels were measured at different time points prior to biopsy/surgery. The authors should further discuss how this could have influenced their results.
10. Authors seem to claim in the first paragraph of the discussion that this study provides evidence for CRP being involved in creating a cancer-friendly microenvironment for ES. Although this can be speculated, this study provides no evidence to support this claim.
11. In the discussion section, the authors should further discuss the advantages/disadvantages of using CRP pretreatment level as a prognostic factor compared to other established prognostic factors. They do a good job describing other studies that examine potential prognostic factors for ES, but how does this study relate to them and what advantages does CRP evaluation have?

Author Response
We would like to thank you for your thorough review and thoughtful comments. We have modified and revised the resubmitted manuscript according to the comments. Please find our point-by-point responses below.
-Simple summary is very similar to abstract. Could be re-worded and simplified for general accessibility.
Adjusted accordingly.
-In simple summary and abstract, “2.9 out of 1,000.000” is written. This should be changed to “2.9 out of 1,000,000” for clarity and consistency with the rest of the article.
Adjusted accordingly.
-Page 2, line 73. “Because of the relationship between inflammation and cancer” is not a complete sentence and should be reworded.
Adjusted accordingly.
-Page 3, lines 119-120. “… was measured By the area under the curve…”. There appears to be an extra space in the sentence and “By” should not be capitalized.
Adjusted accordingly.
-Page 10, line 269. “In view results of this paper,…” should be reworded for clarity.
Adjusted accordingly.
-In Figure 1, the box plot is labelled “DO”. It should be labelled “DOD” for clarity and consistency.
Adjusted accordingly.
-Figures 1-3 are not labelled well. Y-axes are not labelled and the quality of the graphs is relatively low, but readable. All the information is contained in the legends, but it would be better if the graphs could be easily interpreted without having to read the legend.
Labeling was adjusted for better comprehension.
-Some sentences/paragraphs should be combined for clarity. Single sentence paragraphs should be combined. For example, the conclusion could probably be combined into a single paragraph.
Adjusted accordingly.
-CRP levels were measured at different time points prior to biopsy/surgery. The authors should further discuss how this could have influenced their results.
Thank you for pointing this out. A clarification of time of analysis and exclusion criteria was made at line 91-95
-Authors seem to claim in the first paragraph of the discussion that this study provides evidence for CRP being involved in creating a cancer-friendly microenvironment for ES. Although this can be speculated, this study provides no evidence to support this claim.
Thank you for this comment. We agree with you and we adjusted accordingly our thesis.
-In the discussion section, the authors should further discuss the advantages/disadvantages of using CRP pretreatment level as a prognostic factor compared to other established prognostic factors. They do a good job describing other studies that examine potential prognostic factors for ES, but how does this study relate to them and what advantages does CRP evaluation have?
Thank you for pointing this out. A discussion of the advantages of a CRP evaluation was made at line 259-264
To conclude, we feel that the paper significantly improved due to your input. We thank you again for your effort. We hope that our responses are considered satisfactory.
For further questions, we remain at your disposal.
Sincerely,
Dr. S.Consalvo